# Effects of Dual-Task Training on Gait Parameters in Elderly Patients with Mild Dementia

**DOI:** 10.3390/healthcare9111444

**Published:** 2021-10-26

**Authors:** Dong-Kyun Koo, Tae-Su Jang, Jung-Won Kwon

**Affiliations:** 1Department of Physical Therapy, College of Health Sciences, Dankook University, Cheonan 31116, Korea; definikk@gmail.com; 2Department of Medicine, College of Medicine, Dankook University, Cheonan 31116, Korea; jangts@dankook.ac.kr

**Keywords:** mild dementia, gait analysis, dual task

## Abstract

This study aimed to investigate the effectiveness of dual-task training (DTT) compared to single-task training (STT), on gait parameters in elderly patients with mild dementia (MD). Twenty-four elderly patients with MD were randomly assigned to the DTT (*n* = 13) or the STT group (*n* = 11). The DTT group performed a specific cognitive-motor DTT, while the STT group received only motor task training. Both training sessions lasted 8 weeks, with a frequency of 3 days per week, and the cognitive functions and gait parameters were measured. A statistically significant interaction effect was found between the two groups in stride length, stride velocity, cadence, step length, swing phase, stance phase, and double support phase (*p* < 0.05). After 8 weeks, the DTT group showed significant improvement in spatiotemporal parameters, except for the kinematic parameters (*p* < 0.05). In the between-group analysis, the DTT group showed more improvement than the STT group in stride velocity, step length, swing phase, stance phase, and double support (*p* < 0.05). These findings suggest that improvements in spatiotemporal gait parameters after DTT are reported in patients with MD. Our results can guide therapists to include dual tasks in their gait rehabilitation programs for the treatment of mild dementia.

## 1. Introduction

Dementia is a broad term used to describe the symptoms of brain disease that leads to a gradual and long-term decline in cognitive function, including visual perception, problem-solving, and the ability to focus and pay attention. These symptoms lead to physical dysfunction that can affect daily activities [1,2]. Patients with dementia often have gait impairment, which reduces mobility and can lead to the patient experiencing an increase in incidental falling [3]. The course of dementia is generally described in several stages that show the pattern of progressive cognitive and functional impairment: subjective cognitive impairment, mild cognitive impairment, mild dementia, moderate dementia, and severe dementia [4]. Epidemiological studies undertaken in the last ten years have revealed that the prevalence of dementia in the population over the age of 65 is close to 5%. The global population of demented patients is expected to rise from 243 million in 2001 to 81.1 million in 2040 [5].

Mild dementia (MD) is closely associated with cognitive and physical functions, such as learning, memory, balance, and gait [6,7,8]. Cognitive processes affect gait in cognitively normal older adults as well as those with mild cognitive impairment [9]. Impaired cognitive function can lead to decreased walking speed, step length, step frequency, and increased gait variability [10,11]. In addition, gait parameters have a significant correlation with an executive function that supports the role of attention and cognitive function in dual-task gait [12]. Thus, the measurement of gait parameters can be used as an indicator to improve the diagnostic process of dementia [13,14]. Dual-task is defined as the ability to simultaneously perform two tasks, compared to the performance of a single task. The two tasks interfere with each other, and it is assumed that both tasks compete for the same level of information processing resources in the brain [15]. Dual-task decrements in gait parameters were higher in elderly patients with MD compared to elderly patients who did not have MD [16]. Impaired dual-task performance during gait has been correlated with an increased risk of falling, reduced functional abilities, and poor quality of life [17,18]. A previous study reported that dual-task training (DTT) is effective in improving cognitive function by increasing cerebral blood flow [19]. Furthermore, DTT may have a modest impact on spatiotemporal gait parameters and balance in Parkinson’s disease and Alzheimer’s disease [20]. DTT, which includes simultaneously performing motor and cognitive tasks, is suggested as a practical exercise method for MD over single-task training (STT), which includes only motor tasks [21,22]. In addition, meta-analysis research relevant to STT found that there was only limited evidence regarding whether this method of training enhances cognitive function, and its statistical force was low, making it inconclusive [20,23]. Therefore, in the present study we investigated the effectiveness of DTT compared to STT, on spatiotemporal and kinematic gait parameters in elderly patients with MD.

## 2. Materials and Methods

### 2.1. Subjects

The study design was a randomized controlled trial. Twenty-eight elderly patients with MD, hospitalized after diagnosis by a specialist at the S nursing hospital in Gyeongsangbuk-do, were recruited for this study. All subjects were randomly allocated to either the DTT or the STT group. Inclusion criteria for participation were as follows: (1) diagnosis of MD and age over 65 years; (2) Mini-Mental State Examination-Korean version scores between 18 and 23 [24,25,26]; (3) Global Deterioration Scale scores between 4 and 5 [26,27], (4) MD diagnosed at least 6 months prior, and (5) no significant visual, auditory, or vestibular impairment. Subjects were excluded if they were unable to undergo training due to paralysis or severe balance disorders. There was one drop-out in the DTT group and three in the STT group. All the dropouts could no longer participate because of their unstable medical conditions. Therefore, a total of 24 participants were included in the study (Figure 1). Demographic and clinical data were also collected. Ethical approval was obtained from the Institutional Review Board of Daegu University and written informed consent was obtained from all of the participants at enrollment (1040621-201711-HRBR-004-002).

### 2.2. Measurements

(1) Mini-Mental State Examination-Korea version (MMSE-K)

The MMSE-K was used to evaluate the cognitive function and as a dementia screening tool for the elderly. The MMSE-K comprises 12 questions with a total score of 30 points, including items related to memory, attention, calculation, language, understanding, judgment, and time and place orientation. The following cut-off levels were employed to classify the severity of cognitive impairment: 24 points or higher indicated a healthy mental state, 18 to 23 points indicated mild dementia, and 18 points or lower indicated moderate or severe dementia [28]. The MMSE-K is a test for the screening of dementia with the ability to diagnose DSM-IV dementia, measured with the area under the receiver operating characteristics curve at 0.93 [29]. One physical therapist measured and managed MMSE-K.

(2) Global deterioration scale (GDS)

The GDS is a comprehensive assessment of the cognitive, social, and daily life functions of patients with suspected or diagnosed dementia. The overall degeneration scale consists of seven stages that distinguish the severity of the symptoms of dementia: GDS 1, no cognitive impairment; GDS 2, subjective cognitive; GDS 3, mild cognitive impairment; GDS 4, mild dementia; GDS 5, moderate dementia; GDS 6, moderate or severe dementia; and GDS 7, severe dementia. [30]. The GDS reliability is 0.93–1.00, and the validity is 0.80–0.99 [31]. One physical therapist measured and managed GDS.

(3) Gait analysis

The kinematic and spatiotemporal parameters of gait were objectively measured using a validated LEGSys+ wearable device. The inertial measurement unit system is reliable and has been validated against other kinetic and kinematic gold standards [32]. Three-dimensional acceleration and angular velocity of the shins, thighs, and trunk were measured using five wearable sensors. Each Bluetooth-connected sensor included a triaxial accelerometer and a triaxial gyroscope to obtain gait parameters, which were attached to the anterior surface of both the shins 3 cm above the ankle, the anterior surface of both then thighs 3 cm above the knee, and the center of the posterior superior iliac spine. For statistical analysis, the LEGSys+ signal data were sampled at 100 Hz. Gait was assessed with a minimum of five strides in a 7 m walkway. The data were recorded with three strides, excluding the first and last strides. Gait analysis was performed three times under usual conditions to obtain an average value. The data were collected by an experienced physical therapist.

### 2.3. Motor and Cognitive Task Training

DTT requires attention to be paid to both tasks when they are being performed concurrently. Exercise tasks are designed to be as simple as possible so that selective attention can be provided to cognitive tasks. The contents of the task training were established based on previous studies on MD [33,34,35]. Training lasted for a total of 8 weeks, 3 times a week, with each exercise method having a duration of 30 min. The STT group performed only motor tasks, while the DTT group performed motor and cognitive tasks simultaneously. Cognitive tasks were randomly assigned and performed simultaneously with the motor task. Motor tasks consisted of six individual tasks, including shifting from a sitting position to a standing position, reaching out with the arm in various directions while standing, raising the heel while standing, kicking a ball step-by-step, walking on a flat surface, and exercising using a cycle ergometer. Cognitive tasks consisted of five individual tasks, including counting backward, performing arithmetic, word association test, picture matching task, and remembering the picture (Table 1) [33,36].

### 2.4. Experimental Procedure

During the motor learning stage [37], both groups performed only motor task training for two weeks. At the beginning of the 3rd week, the DTT group performed both executive and cognitive tasks simultaneously. Cognitive function and gait parameters were assessed in both groups before and after the training. When a subject was unable to sustain training due to limited physical ability, training was stopped for that subject. If a subject expressed fatigue, pain, or abnormal breathing, the subject was allowed a 5 min break.

### 2.5. Statistical Analysis

SPSS Statistics (version 25) (IBM, Armonk, NY, USA) for Windows was used for data analysis. A Shapiro-Wilk test was conducted to determine the normal distribution of each parameter, and all parameters were normally distributed except the height parameter. The chi-squared test and independent *t*-test were performed to analyze the differences between the two groups in terms of sex, age, weight, MMSE-K, and GDS. The height variable was analyzed by the Mann-Whitney test. Two-way repeated-measures analysis of variance was used to compare gait parameters before and after training between the DTT and STT groups. The paired *t*-test was then used to compare gait parameters before and after training in each group. Null hypotheses of no difference were rejected if *p*-values were less than 0.05. Results are presented as the mean ± standard deviation or n (%). Clinical and demographic data of dropouts were excluded.

## 3. Results

In the demographic data, no significant differences were observed in terms of age, sex, height, weight, MMSE-K, and GDS scores between the DTT and STT groups (*p* > 0.05; Table 2). Table 3 shows the spatiotemporal and kinematic parameters of gait before and after training. No significant differences in gait parameters were observed between the two groups in the pre-test (*p* > 0.05). In the post-test, a statistically significant interaction effect was found between the two groups in stride length, stride velocity, cadence, step length, swing phase, stance phase, and double support phase (*p* < 0.05). There was a significant main effect seen on the spatiotemporal gait parameters (*p* < 0.05), however, there was no significant main effect observed for kinematic gait parameters (*p* > 0.05). In the DTT group, stride length, stride velocity, cadence, step length, and swing phase significantly increased (*p* < 0.05). In addition, stride time, stance phase, and double support phase significantly decreased after training (*p* < 0.05), but there were no significant differences in the knee and the hip angles (*p* > 0.05). In the STT group, there were no significant differences in spatiotemporal and kinematic gait parameters after training.

## 4. Discussion

In the present study, we investigated the clinical effectiveness of DTT in patients with MD. We compared the efficacy of DTT on the spatiotemporal and kinematic gait parameters. We found that stride length, stride velocity, cadence, step length, swing phase, stance phase, and double support phase significantly improved in the DTT group compared to the STT group. These results suggest that DTT is more beneficial than STT in improving gait function in elderly patients with MD and that its superiority is due to increased executive function.

Dual-task performance could have more significant effects due to of its higher task automatization and more efficient integration of task-related networks [38]. Furthermore, improvements in gait parameters may result in an improved ability to prioritize a motor task over a cognitive task. Due to limited central processing resources, the subject can prioritize less complex tasks [39]. In the present study, we found that DTT could improve spatiotemporal gait parameters by improving gait automatization and reducing competition for attentional resources among the tasks. In our study, significant improvements in spatiotemporal parameters were observed after DTT treatment. Previous studies suggested that DTT could improve gait velocity and other spatiotemporal parameters, such as stride time and double-support percentage. This indicated that DTT programs improved gait automatization and decreased competition for attentional resources between tasks [38,40].

There was no significant difference in joint angles before and after STT and DTT. In a previous study, the kinematic and kinetic variables were less affected in older adults than in young adults or children because the variations were found only at fast speeds, whereas the ground reaction forces did not change with the speed [41] To affect kinematic variables in the elderly, a fast walking speed is required, therefore, we believe that this study did not show a significant difference in joint variables due to the slow walking speed of the elderly [42]. In addition, interventions designed to enhance the range of motion have had little impact on gait characteristics in older adults, especially with regards to joint motion during gait [43]. Based on previous studies, we believe that the training methods performed in the present study did not significantly change the characteristics of the joint angle in elderly patients with dementia. In the STT group, spatiotemporal and kinematic parameters were not significantly different after training. A previous study reported that single-task programs were not superior to dual-tasks in improving gait function, and subjects who received DTT showed greater improvements in single-task gait speed [44] In this study, the relatively short intervention period of 8 weeks may account for the lack of significant improvement.

When comparing the mean differences between DTT and STT, significant differences were found in stride length, stride velocity, cadence, step length, swing phase, stance phase, and double support phase. Previous studies reported that DTT was effective in improving gait performance, motor symptoms, and balance in Parkinson’s disease [45]. and in older adults with early dementia, compared to other training or non-interventional methods [46]. Moreover, DTT improves attention and memory in elderly people with mild cognitive impairment and it also improves their overall cognitive function [47]. Physical activity has been shown to enhance neuroplasticity by elucidating the cellular and molecular mechanisms [48] and the interconnections [49] associated with increased neurotrophins, particularly brain-derived neurotrophic factors. These exercise-induced changes in brain regions may explain the effects of DTT on executive functions, although it remains to be seen whether these changes can be modified when motor training is performed concurrently with cognitive tasks. Consequently, DTT may be beneficial for patients with MD through multiple pathways by inducing improved gait parameters.

The results of this study suggest that cognitive-motor DTT may be incorporated into MD rehabilitation protocols to improve gait function. However, this study had several limitations. First, the sample size was relatively small and included individuals classified between 20 to 23 points of the MMSE-K score, excluding those with an MMSE-K score of <20, this was done because it was difficult for subjects with severe dementia to perform our task training program. Therefore, the sample size may not be representative of the entire population of patients having dementia. Second, we used standardized task training as a way of standardization. Therefore, the training program was not designed to suit the abilities of each subject. Future studies are required to address these limitations. Third, this study analyzed only the dual-task training effect compared to general intervention. Future research needs to clearly analyze the data by adding sham condition. Fourth, since this study did not measure the appropriate exercise intensity for each patient, future study needs to analyze cardiopulmonary functions. Fifth, this study needs to specifically analyze the absence of imaging parameters and aspects of cognitive deficits.

In conclusion, improvements in spatiotemporal gait parameters after DTT have been reported in patients with MD. We believe that DTT can help manage patients with MD. In particular, the effect of DTT on spatiotemporal gait parameters can be an important issue when planning interventional strategies for slowing down the progression of MD.

## Figures and Tables

**Figure 1 healthcare-09-01444-f001:**
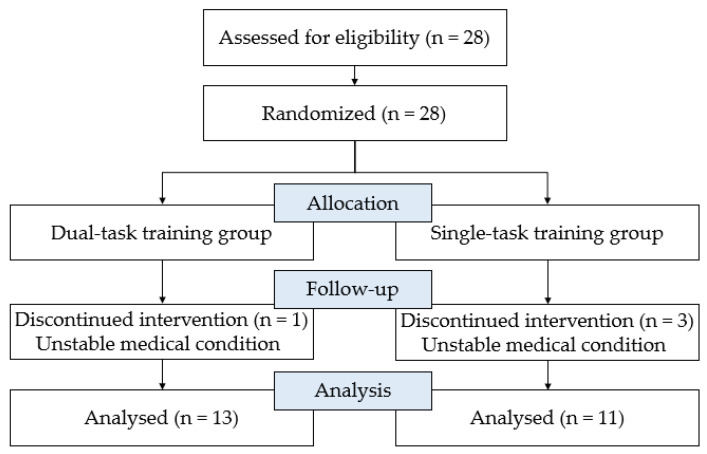
Flow diagram for the study.

**Table 1 healthcare-09-01444-t001:** Dual-task training methods.

Motor Training Tasks	Cognitive Training Tasks
Sit to standing (5 min) Reaching arm in various directions on standing (5 min) Raising the heel on standing (5 min) Kicking a ball step by step (5 min) Walking on flatland (5 min) Exercising cycle ergometer (5 min)	(1) Counting backward (ex) 100, 99, 98… (2) Doing arithmetic (ex) 2 + 5, 9 − 4… (3) Word association test (ex) Say a word starting with a consonant (4) Picture matching task (ex) things, places, animals… (5) Remember and answer (ex) Pictures, letters, objects…

**Table 2 healthcare-09-01444-t002:** General characteristics.

	DTT Group	STT Group	x^2^/t/z	*p*
Sex (M/F)	5(38%)/8(62%)	4(36%)/7(64%)	0.011	0.920
Age (years)	80.92 ± 8.40	76.23 ± 6.37	1.521	0.143
Height (cm)	154.16 ± 10.54	154.94 ± 9.56	−0.345	0.520
Weight (kg)	58.65 ± 9.26	53.50 ± 14.27	0.922	0.367
MMSE-K (score)	20.69 ± 2.10	20.27 ± 1.90	0.509	0.616
GDS (score)	4.23 ± 0.73	4.33 ± 1.15	0.259	0.387

Values represent mean ± standard deviation; DTT, dual-task training; STT, single-task training; MMSE-K, mini mental state examination-korean version; GDS, global deterioration scale.

**Table 3 healthcare-09-01444-t003:** The data for the gait parameters before and after training between the groups.

	DTT				STT				F (Time)	F (Time × Group)
	Pre-Test	Post-Test	t	ES	Pre-Test	Post-Test	t	ES		
Stride time (s)	1.52 (0.40)	1.26 (0.18)	−3.35 ^†^	0.84	1.31 (0.32)	1.23 (0.19)	−1.06	0.30	9.94 *	3.13
Stride length (m)	0.66 (0.11)	0.77 (0.10)	3.38 ^†^	1.05	0.76 (0.28)	0.77 (0.25)	0.32	0.04	6.73 *	4.60 *
Stride velocity (m/s)	0.45 (0.09)	0.62 (0.13)	6.48 ^†^	1.52	0.61 (0.28)	0.62 (0.20)	0.14	0.04	12.42 *	10.68 *
Cadence (steps/min)	83.52 (18.74)	97.21 (12.74)	3.90 ^†^	0.85	98.23 (26.50)	99.95 (21.83)	0.42	0.07	8.24 *	4.98 *
Step length (m)	0.32 (0.06)	0.40 (0.04)	4.85 ^†^	1.57	0.38 (0.10)	0.38 (0.08)	−0.61	<0.01	9.67 *	13.70 *
Swing (%)	33.30 (3.08)	38.14 (2.57)	7.05 ^†^	1.71	36.14 (4.36)	36.67 (4.02)	0.53	0.13	20.74 *	13.45 *
Stance (%)	66.70 (3.08)	61.86 (2.57)	−7.05 ^†^	1.71	63.86 (4.36)	63.34 (4.02)	−0.53	0.12	20.74 *	13.44 *
Double support (%)	33.41 (6.16)	23.73 (5.14)	−7.05 ^†^	1.71	27.72 (8.72)	26.67 (8.04)	−0.53	0.13	20.75 *	13.43 *
Left knee ROM (°)	38.66 (5.37)	41.70 (7.59)	2.06	0.46	41.42 (11.51)	41.95 (17.37)	0.21	0.04	1.64	0.82
Right knee ROM (°)	35.75 (7.95)	37.57 (9.60)	0.79	0.21	39.23 (12.36)	42.77 (15.70)	1.17	0.25	2.07	0.21
Left hip ROM (°)	31.26 (5.28)	33.39 (4.69)	1.02	0.43	32.45 (10.59)	32.48 (8.41)	0.02	<0.01	0.57	0.54
Right hip ROM (°)	32.77 (5.49)	34.02 (6.02)	0.46	0.22	32.34 (10.42)	33.71 (8.61)	0.60	0.14	<0.01	<0.01

Values represent mean (± standard deviation); DTT, dual task training group; STT, single task training group; ROM, range of motion; ES, effect size; * *p* < 0.05; Significant difference within group; ^†^
*p* < 0.05.

## Data Availability

The data presented in this study are available on request from the corresponding author.

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
