# Peer review of "Effects of Dual-Task Training on Gait Parameters in Elderly Patients with Mild Dementia"

_healthcare, 2021, doi:10.3390/healthcare9111444_

Round 1

Reviewer 1 Report

In this article, the Authors describe an experimental approach to cognitive rehabilitation using dual-task training in a group of elderly patients with dementia, since DT a concept of real interest in the modern era of neurology. However, the article suffers from several issues. First, Table 3 – albeit cited in the article – is missing, and it is not possible to analyse gait parameters. Second, the methods, inclusion criteria and statistical analysis are very unclear. For example, the authors state that they included only individuals with mild dementia, but in the inclusion criteria the MMSE range is 18-23, and in table 2 there are clearly MMSE and GDS values compatible with the diagnosis of moderate dementia, and the mean is 20 points. Finally, contradicting the methods sections, the authors said that they have excluded patients with MMSE of 19 or below, but data say the opposite (20 minus 2SD is 18, 18.5 at best).

Other points are:

  1. Introduction, lines 31-33: I suggest adding ‘subjective cognitive impairment’ when describing the dementia spectrum. Moreover, epidemiological data and considerations are missing in the introduction.
  2. Methods: how many patients were screened? Who administered the MMSE and calculated the GDS?
  3. Methods, intervention: were the subjects blinded to their allocation? Why you did not choose a sham comparison? It is difficult-impossible for DT?
  4. Statistical analysis: do any power analyses were performed? How do you test for normality? Do demographical and clinical data of the drop-outs were compared with dataof retained patients?
  5. Statistical analysis: since the groups are relatively small, I am doubtful that the variables follow a normal distribution. If it is, authors should specify it by adding the values of the shapiro-wilk test or other normality tests. If that is not the case, I suggest using the mann-whithney t-test or the wilcoxon signed-rank test or the Mcnemar test when appropriate to compare the median rather than the mean.
  6. Table 1: I suggest adding the cognitive abilities tapped by each cognitive task, and if the exercise was aerobic or not.
  7. Do you used any methods to measure the exercise intensity (i.e. hearth rate, perceived fatigue)?
  8. Discussion, 172-175: This sounds as a repetition of the previous paragraph, please rewrite. Moreover, in this section the authors say 'precentage' regarding double support and swing phase, but this parameter is not specified in the methods section. Please rewrite.
  9. Discussion, 179-186: I suggest moving this paragraph (at least in part) earlier in the discussion section, which has to be organized by first discussing the available literature and then the results of the present study by accurately comparing them with the literature.
  10. Discussion, 219-226: I would add the absence of imaging parameters and of a further characterization of cognitive deficits as a limitation, next to the absence of a sham rehabilitation group as control, and the absence of randomization and blinding procedures.

Reviewer 2 Report

Dear authors

Thank you for the opportunity to review your article.

Brief summary: This is a prospective, randomized  study that aimed to investigate the effectiveness of dual-task training (DTT) compared to 9 single-task training (STT), on gait parameters in elderly patients with mild dementia (MD). The findings have important practical implications that can be introduced to improve the quality of life of these patients.

Areas of strength

The references included are relevant for the subject under study show 14/46(30%) references from the last 5 years. There is strong concordance between the and the methods used. The description of the methodology was made in a clear and adequate way, although the designation of the study design and the CONSORT flow diagram are missing. the results are clearly described, however, table 3 is missing. The discussion correlates with the presented data and takes the published literature into account. The manuscript presents some limitations and clinical implications.

Weakness:

  1. Page 2, line 56 – correct please – “(…) gait parameters in elderly patients with MD.” To “ (…) spatiotemporal and kinematic gait parameters in elderly patients with MD.”

  1. Page 2, line 57. Missing Study design. Please define the study design. Prospective, randomized, non-blind, two-arm study? The manuscript has been written following CONSORT guidelines for non-pharmacological? Please report objectively if you used the CONSORT guidelines.

  1. Page 2, line 72 - missing Flow diagram CONSORT

  1. Page 3, line 143 – correct please “Results are presented as the mean ± standard deviation” to “Results are presented as the mean ± standard deviation or n(%).”

  1. Page 3, line 147 - Table 3 is missing.

  1. Page 4, table 2 – please put n(%).

Sex (M/F)

5 (38%)/8 (62%)

4(36%) /7(64%)

  1. Page 5 line 181-183 Furthermore, the improvement in gait parameters 186 may be due to the enhanced ability to prioritize a motor task over a cognitive task.”

 Is the same on line 186-187 “Furthermore, the improvement in gait parameters 186 may be due to the enhanced ability to prioritize a motor task over a cognitive task.”

  1. The references do not follow the journal guidelines, the journal name is missing and the year must be written in bold.

Round 2

Reviewer 2 Report

Dear authors

Thanks for the opportunity to review the article again. Authors improved the manuscript and its quality increased. It has conditions to be accepted.